# Prognostic Value of Inflammation Scores and Hematological Indices in IgA and Membranous Nephropathies: An Exploratory Study

**DOI:** 10.3390/medicina60081191

**Published:** 2024-07-23

**Authors:** Nicolae Pană, Gabriel Ștefan, Tudor Popa, Otilia Ciurea, Simona Hildegard Stancu, Cristina Căpușă

**Affiliations:** 1Department of nephrology, “Carol Davila” University of Medicine and Pharmacy, 050474 Bucharest, Romania; 2“Diaverum Morarilor” Nephrology and Dialysis Medical Center, 022452 Bucharest, Romania; 3“Dr. Carol Davila” Teaching Hospital of Nephrology, 010731 Bucharest, Romania

**Keywords:** systemic-inflammation-based prognostic scores, hematological indices, membranous nephropathy, IgA nephropathy, end-stage kidney disease, mortality

## Abstract

*Background and Objectives:* Systemic-inflammation-based prognostic scores and hematological indices have shown value in predicting outcomes in various clinical settings. However, their effectiveness in predicting outcomes specifically for IgA nephropathy (IgAN) and membranous nephropathy (MN), the most common primary glomerular diseases diagnosed by kidney biopsy, has not been thoroughly investigated. *Materials and Methods:* We conducted a retrospective, observational study involving 334 adult patients with biopsy-proven IgAN (196 patients) and MN (138 patients) from January 2008 to December 2017 at a tertiary center. We assessed six prognostic scores [Glasgow prognostic score (GPS), modified GPS (mGPS), prognostic nutritional index (PNI), neutrophil-to-lymphocyte ratio (NLR), platelet-to-lymphocyte ratio (PLR), lymphocyte-to-C-reactive protein ratio (LCRP)] and two hematological indices [red blood cell distribution width (RDW), platelet distribution width (PDW)] at diagnosis and examined their relationship with kidney and patient survival. *Results:* End-stage kidney disease (ESKD) occurred more frequently in the IgAN group compared to the MN group (37% vs. 12%, *p* = 0.001). The mean kidney survival time was 10.7 years in the IgAN cohort and 13.8 years in the MN cohort. After adjusting for eGFR and proteinuria, lower NLR and higher LCRP were significant risk factors for ESKD in IgAN. In the MN cohort, no systemic-inflammation-based scores or hematological indices were associated with kidney survival. There were 38 deaths (19%) in the IgAN group and 29 deaths (21%) in the MN group, showing no significant difference in mortality rates. The mean survival time was 13.4 years for the IgAN group and 12.7 years for the MN group. In the IgAN group, a lower PLR was associated with a higher mortality after adjusting for age, the Charlson comorbidity score, eGFR, and proteinuria. In patients with MN, higher NLR, PLR, and RDW were associated with increased mortality. *Conclusions:* NLR and LCRP are significant predictors of ESKD in IgAN, while PLR is linked to increased mortality. In MN, NLR, PLR, and RDW are predictors of mortality but not kidney survival. These findings underscore the need for disease-specific biomarkers and indicate that systemic inflammatory responses play varying roles in the progression and outcomes of these glomerular diseases. Future studies on larger cohorts are necessary to validate these markers.

## 1. Introduction

IgA nephropathy (IgAN) and membranous nephropathy (MN) are the most common primary immune-mediated glomerular diseases diagnosed through kidney biopsy, each having a highly variable prognosis [1].

IgAN is characterized by the deposition of IgA immune complexes in the glomeruli, leading to local inflammation and glomerular injury. This condition often presents with hematuria and varying degrees of proteinuria. The current markers for IgAN include serum creatinine, proteinuria levels, and eGFR, though these are not specific to the disease’s unique immune-mediated mechanisms. MN involves the deposition of immune complexes on the glomerular basement membrane, causing thickening and damage to the glomeruli. Patients typically present with heavy proteinuria and nephrotic syndrome. The established markers include anti-PLA2R antibodies and proteinuria levels, which are used in clinical practice to monitor the disease activity and response to therapy.

The current guidelines recommend tailoring treatment based on the level of immunologic activity and overall prognosis [1]. However, using proteinuria and the glomerular filtration rate as the primary indicators for the prognosis and treatment of primary glomerular diseases can be misleading, as these metrics can be affected by chronic damage and nephron scarring, which are unrelated to immune activity. Therefore, there is ongoing effort to identify affordable and readily available markers that accurately reflect immune activity in these glomerular diseases.

A complete blood count (CBC) test, commonly ordered in clinical practice, measures key blood components like white blood cells (WBCs), neutrophils (NEs), lymphocytes (LYs), and platelets (PLTs). This test provides a general view of the immune system, which is crucial for diagnosing and managing patients. NE and LY are vital for innate and adaptive immunity, respectively. Increased NE indicates inflammation, while decreased LY suggests poor nutrition and prognosis [2]. The neutrophil-to-lymphocyte ratio (NLR) combines these markers to evaluate systemic inflammation. Platelets also play a role in inflammation, leading to the use of the platelet-to-lymphocyte ratio (PLR) as a marker of acute inflammatory and prothrombotic conditions in various diseases, including rheumatic and cardiovascular conditions [2]. Both NLR and PLR have shown prognostic value in coronary artery disease, solid tumors, rheumatoid arthritis, chronic kidney disease (CKD), acute kidney injury (AKI), and rapidly progressive glomerulonephritis [2,3].

Similarly, red cell distribution width (RDW) and platelet distribution width (PDW) are routinely reported in complete blood counts and serve as important prognostic biomarkers [4]. Higher RDW within the normal range is associated with cardiovascular outcomes like myocardial infarction, heart failure, and stroke, which is potentially due to elevated proinflammatory substances affecting erythropoiesis and vascular health [5,6]. The Glasgow prognostic score (GPS), which includes C-reactive protein (CRP) and serum albumin, is a concise tool for assessing systemic inflammation and nutritional status. Extensively validated in oncology, the GPS is also linked to CKD progression and mortality [7,8,9]. However, there are limited data on the value of systemic-inflammation-based prognostic scores and hematological indices of inflammation in IgAN and MN.

In this exploratory study, we aimed to investigate the relationship between various peripheral blood cell scores and indices with kidney and patient survival outcomes in IgAN and MN. We selected six prognostic scores [Glasgow prognostic score (GPS), modified GPS (mGPS), prognostic nutritional index (PNI), NLR, PLR, and lymphocyte-to-C-reactive protein ratio (LCRP)] and two hematological indices [red blood cell distribution width (RDW) and platelet distribution width (PDW)] at diagnosis because they have been previously validated in different nephrological scenarios. Our goal was to enhance the understanding of their predictive value for critical outcomes, specifically end-stage kidney disease (ESKD) and mortality, in these common glomerular diseases.

## 2. Materials and Methods

### 2.1. Study Population

We conducted a retrospective, observational study of all adults with biopsy-proven IgAN and MN from January 2008 to December 2017 in the nephrology department of the “Dr. Carol Davila” Teaching Hospital of Nephrology, Bucharest, Romania.

From a total of 472 native kidney biopsies at our center with the diagnosis of IgAN and MN, we excluded patients who underwent more than one biopsy during the study period (11 patients—5 with IgAN, 6 with MN), those with solid tumors (6 patients—3 with IgAN, 3 with MN), those with active infections (20 patients—17 with IgAN and 3 with MN), those who underwent biopsy after beginning kidney replacement therapy (1 patient), those who received immunosuppressive treatment prior to the kidney biopsy (37 patients), and those with insufficient material for a complete histopathologic analysis (32 patients) or insufficient clinical data (31 patients). The final cohort for analysis comprised 334 patients—196 with IgAN and 138 with MN.

This study was conducted with the provisions of the Declaration of Helsinki and was approved by the local ethical committee (“Dr. Carol Davila” Teaching Hospital of Nephrology, Bucharest, Romania, approval number 40/04.08.2023).

### 2.2. Clinical Data

Patient demographics and clinical data were retrieved from electronic medical records using a standardized data form. Two analysts conducted the database search to ensure accuracy and reliability, resolving discrepancies through consensus to ensure a rigorous review process.

The collected data included age, Charlson comorbidity index, presence of arterial hypertension (defined as blood pressure >140/90 mmHg or the use of antihypertensive agents), complete blood count, inflammation markers (C-reactive protein, erythrocyte sedimentation rate, fibrinogen), serum albumin, eGFR calculated by the four-variable MDRD formula, proteinuria (quantified by 24-h urine collection and expressed as urinary protein-to-creatinine ratio, in g/g), and hematuria (red blood cells per high power field). Only data at the time of the biopsy were used in the analyses.

Clinical syndromes were retrospectively defined as nephrotic syndrome, nephritic syndrome, CKD, AKI, and asymptomatic urinary abnormalities (AUAs).

The selected six prognostic scores—GPS, mGPS, PNI, NLR, PLR, and LCRP—and two hematological indices—RDW and PDW—were defined according to the definitions provided in Table 1.

### 2.3. Study Endpoints

The primary endpoint of this study was the incidence of ESKD. This was determined by the need to initiate dialysis or undergo a kidney transplantation in the patient cohort. A patient was considered to have reached the primary endpoint when either of these events was registered in the Romanian Renal Registry.

The secondary endpoint of this study was all-cause mortality. This was measured by tracking the number of deaths, regardless of the cause, in the patient cohort during the follow-up period.

All patients were followed from the start of the study (id est, the moment of kidney biopsy) until 30 April 2024.

### 2.4. Statistical Analysis

Continuous variables are presented as mean or median with 95% confidence interval (CI) after testing normality with the Shapiro–Wilk test, while categorical variables are presented as percentages. To evaluate differences between IgAN and MN, Student’s *t*-test or the Mann–Whitney test was applied, depending on the distribution of the variables. For categorical variables, the chi-square test was utilized to determine statistical significance.

The Kaplan—Meier method was employed to analyze the probability of event-free survival, with the log-rank test used for comparisons between IgAN and MN. To identify independent predictors—systemic-inflammation-based prognostic scores and hematological indices—contributing to the progression to ESKD and mortality, both univariate and multivariate analyses (based on the Cox proportional hazard ratio) were performed for each glomerular disease. In the ESKD models, the inflammatory scores were adjusted for eGFR and proteinuria, and, in the mortality models, adjustments were made for eGFR, proteinuria, and Charlson comorbidity score. The results are expressed in terms of hazard ratios (HRs) with a 95% CI. We employed two methods to test for collinearity among our predictor variables: (i) the variance inflation factor (VIF), with a desirable threshold of VIF < 10, and (ii) the absolute value of correlation coefficients, aiming for |r| or |rs| < 0.7. Our analysis indicated no significant collinearity among the variables used in the Cox proportional hazard models.

Statistical analyses were performed using SPSS (IBM SPSS Statistics for Macintosh, Version 27.0. Armonk, NY, USA: IBM Corp) and GraphPad Prism (GraphPad Software (Version 9.5.1 (528)), La Jolla, CA, USA).

## 3. Results

### 3.1. Baseline Demographic and Clinical Characteristics

The baseline demographic and clinical characteristics of the cohort are presented in Table 2. Patients with MN were generally older than those with IgAN (55 versus 43 years). As expected, patients with MN more frequently presented with nephrotic syndrome (85% versus 6%), while those with IgAN more often had nephritic syndrome (85 versus 10%). In line with this, arterial hypertension was more common in patients with IgAN, although the Charlson comorbidity score was similar between the two groups.

At diagnosis, the patients with IgAN had lower eGFR (39.1 versus 65.0 mL/min), higher hematuria (180 versus 20 RBC/HPF), and lower proteinuria (1.2 versus 5.7 g/g) than patients with MN, reflecting their clinical presentation.

In terms of inflammation parameters, there were no significant differences in hemoglobin levels, neutrophil count, lymphocyte count, or C-reactive protein levels between the two groups. However, patients with MN had higher platelet counts, erythrocyte sedimentation rates, fibrinogen levels, and lower serum albumin levels.

Immunosuppressive treatment was more commonly received by patients with MN, whereas ACEI/ARB usage was similar between the two groups.

The median follow-up time for the entire cohort was 113.0 months (95% CI 108.5–117.4).

### 3.2. Systemic-Inflammation-Based Prognostic Scores and Hematological Indices

According to the GPS scoring system, including the modified form, the patients with MN had higher GPS scores. Similarly, the PNI was also higher in the MN cohort.

There were no differences in the NLR and LCRP between the glomerular diseases; however, patients with MN had a higher platelet-to-lymphocyte ratio.

Regarding hematological indices, the patients with MN had a higher RDW, while the PDW was similar between the two groups.

### 3.3. Kidney Survival

The ESKD endpoint occurred more frequently in the IgAN group than in the MN group (37% versus 12%, *p* = 0.001).

In the IgAN cohort, the mean kidney survival time was 10.7 years (95% CI: 9.8–11.7). Renal survival rates at 1, 3, 5, and 10 years were 92%, 84%, 73%, and 60%, respectively. In the MN group, the mean kidney survival time was 13.8 years (95% CI: 13.0–14.5). The renal survival rates at 1, 3, 5, and 10 years were 99%, 95%, 91%, and 85%, respectively (Figure 1A).

In univariate Cox regression analysis, GPS and PNI were associated with kidney survival in patients with IgAN. However, after adjusting for eGFR and proteinuria, only a lower NLR and a higher LCRP remained significant risk factors for ESKD (Table 3).

In the MN cohort, none of the systemic-inflammation-based prognostic scores or hematological indices studied were associated with kidney survival in either univariate or multivariate analysis (Table 3).

### 3.4. Patient Survival

A total of 38 deaths (19%) occurred in the IgAN group compared to 29 deaths (21%) in the MN group, indicating no significant difference in the mortality rates between the two glomerular diseases. The primary cause of mortality in the IgAN group was cardiovascular diseases (60%), followed by infectious (20%), gastroenterological (13%), and neoplastic (9%) diseases. Similarly, in the MN group, cardiovascular diseases (45%) and infectious diseases (26%) were the main causes of death.

In the IgAN group, the mean survival time was 13.4 years (95% CI: 12.6–14.1), with survival rates at 1, 3, 5, and 10 years of 94%, 85%, 83%, and 82%, respectively. In the MN group, the mean survival time was 12.7 years (95% CI: 11.8–13.8), with survival rates at 1, 3, 5, and 10 years of 91%, 90%, 87%, and 77%, respectively (Figure 1B).

In the univariate Cox regression analysis for the IgAN group, GPS, mGPS, PNI, PLR, and PDW were associated with mortality. However, after adjusting for age, the Charlson comorbidity score, eGFR, and proteinuria, only a lower PLR was associated with a higher risk of mortality (Table 4). In patients with MN, PNI, NLR, PLR, LCRP, and RDW were associated with mortality in the univariate Cox regression analysis. However, in the multivariate analysis, only higher NLR, PLR, and RDW were associated with mortality (Table 4).

## 4. Discussion

In the present study, we aimed to explore the relationship between common systemic-inflammation-based prognostic scores and hematological indices at diagnosis with long-term outcomes—specifically kidney and patient survival—in IgAN and MN, which are the most frequent glomerular diseases diagnosed by kidney biopsy. The main findings of the study are as follows: (i) In IgAN, NLR and LCRP were identified as risk factors for progression to ESKD. Additionally, PLR was associated with increased mortality. (ii) In MN, there were no systemic-inflammation-based prognostic scores or hematological indices that were associated with kidney survival. However, both NLR and PLR were associated with increased mortality in patients with MN.

### 4.1. Kidney Survival

Previous studies from China identified a higher NLR at diagnosis as an independent risk factor for ESKD in IgAN; however, the median eGFR at baseline in these studies was between 80 and 96 mL/min [16,17,18]. In contrast, in our European population, the median eGFR was 39 mL/min. In our multivariate analysis adjusted for eGFR and proteinuria, a lower NLR emerged as an independent risk factor for ESKD. These findings could be explained by the lower eGFR at diagnosis in our population. In this context, a lower NLR might reflect chronic inflammation, immune exhaustion, and malnutrition. Thus, in earlier stages of IgAN, a higher NLR could reflect the “disease-specific” immune-mediated nephron loss, while, in advanced stages, a lower NLR might reflect the generic response to nephron loss (id est, CKD progression).

While a higher NLR typically reflects acute inflammation (high neutrophil counts), a lower NLR might reflect a shift toward chronic inflammation and immune dysregulation, which are detrimental in long-term IgAN progression [19]. This scenario is consistent with the finding that a higher LCRP (higher lymphocyte counts relative to CRP) was also associated with ESKD.

In our study, none of the systemic-inflammation-based prognostic scores or hematological indices were associated with ESKD in patients with membranous nephropathy (MN). This finding suggests that the progression to ESKD in MN may be influenced more by other factors beyond systemic inflammation and hematological parameters [20]. MN is primarily characterized by immune complex deposition in the glomeruli, leading to podocyte damage and proteinuria, which may not be directly reflected in the systemic inflammatory markers measured [20]. Moreover, the progression of MN can be heavily influenced by other factors such as the degree of proteinuria, response to immunosuppressive therapy, and underlying comorbidities like hypertension and diabetes [21]. These factors may have a more significant impact on renal outcomes in MN than the systemic inflammatory response, which is why the prognostic scores and hematological indices based on inflammation did not show a significant association with ESKD in this patient group. This underscores the need for more specific biomarkers or indicators that directly reflect the pathophysiological mechanisms of MN.

### 4.2. Patient Survival

In relation to the PLR, there are not enough data to define a definitive cut-off point to predict mortality. Patients with IgAN had a significantly lower PLR than patients with MN. Moreover, in the multivariate survival analysis, a lower PLR value was associated with mortality in the IgAN group, while a higher PLR value was associated with mortality in the MN group. These findings suggest a J-curve pattern for death in relation to PLR, which might also reflect the different pathophysiological mechanisms underlying the two diseases.

Specifically, IgAN is characterized by chronic immune-mediated inflammation, where lower PLR values indicate severe disease and immune dysregulation. This is because IgAN involves the deposition of IgA immune complexes in the glomeruli, leading to local inflammation and increased lymphocyte activity, which can reduce the PLR. On the other hand, MN is driven by autoimmune processes targeting glomerular podocytes, resulting in a prothrombotic state. The higher PLR values in MN reflect increased platelet activation and aggregation, which are associated with a higher risk of thrombosis and systemic inflammation, thereby increasing the mortality risk [22].

Supporting our findings, a recent study of over 100,000 incident patients on hemodialysis established a J-curve pattern for mortality in relation to PLR: patients with PLR values below 100 and above 300 had higher mortality rates compared to those with PLR values in the range of 100–150 s [23].

In addition to PLR, a higher NLR was also associated with mortality in patients with MN. Studies suggest that both a high baseline NLR and increases in NLR over time are linked to higher mortality rates. Some authors proposed a cut-off point of NLR ≥ 3.5, beyond which the risk of mortality significantly increases [24]. The strong association between high NLR and low serum albumin levels further supports using NLR as a reliable mortality marker. Therefore, in patients with MN presenting with nephrotic syndrome, NLR may have greater predictive utility than albumin because NLR increases in the blood much faster (6–8 h) compared to the slower decrease in albumin levels (19–21 days) [23,25].

The RDW level may be a useful predictive biomarker for estimating the response to immunosuppressive therapy and reflecting an increased inflammatory response in nephrotic syndrome (NS). Turgutalep et al. found that the RDW level is a powerful and independent predictor of treatment response in patients with NS caused by primary glomerular diseases, including MN [26]. Patients with low or normal RDW levels had a high response rate. Similarly, our study showed that a higher RDW level was associated with increased mortality, further supporting the use of RDW as a prognostic marker in MN.

### 4.3. Limitations

Our study has several limitations that should be taken into account when interpreting the findings. Firstly, being a single-center observational study, the scope and generalizability of our results are inherently limited. The specific patient population and treatment practices at our center may not represent the broader global patient population or the practices at other healthcare centers. Therefore, our results may not be universally applicable to all patients with IgAN or MN.

Secondly, we only analyzed patients with IgAN and MN at the time of presentation, i.e., at the time of kidney biopsy. This means that our findings do not consider potential changes in disease status or treatment responses over time, which could affect long-term outcomes. Moreover, while the chosen confounders are pertinent, other potential confounders (e.g., medication adherence and lifestyle factors, such as diet and physical activity) were not accounted for, which could influence the study outcomes.

Additionally, our study’s sample size restricted our ability to conduct in-depth subgroup analyses for each systemic-inflammation-based prognostic score and hematological index. This highlights the need for larger future studies to comprehensively address these subgroup variations.

## 5. Conclusions

This study highlights the importance of systemic-inflammation-based scores in predicting the outcomes of IgAN and MN. In IgAN, NLR and LCRP are significant predictors for ESKD progression, while PLR is associated with increased mortality. In MN, NLR, PLR, and RDW predict mortality but not kidney survival. These findings underscore the necessity of disease-specific biomarkers and indicate that systemic inflammatory responses influence the progression and outcomes of these glomerular diseases differently. Future studies with larger cohorts are essential for validating these markers and investigating the mechanisms that affect them.

## Figures and Tables

**Figure 1 medicina-60-01191-f001:**
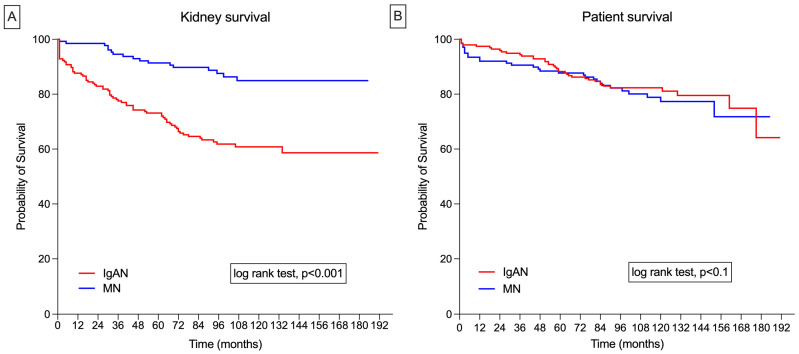
(**A**) Nonadjusted Kaplan–Meier survival curves for IgA nephropathy (IgAN) and membranous nephropathy (MN) for kidney survival; (**B**) nonadjusted Kaplan–Meier survival curves for IgAN and MN for patient survival.

**Table 1 medicina-60-01191-t001:** Inflammation-based scores and hematological markers of inflammation definitions.

Inflammation-Based Score	Definition
Glasgow prognostic score [10,11]	CRP ≤ 10 mg/L and albumin ≥ 3.5 g/dL (0) CRP > 10 mg/L or albumin < 3.5 g/dL (1) CRP > 10 mg/L and albumin < 3.5 g/dL (2)
Modified Glasgow prognostic score [10,11]	CRP ≤10 mg/L (0) CRP > 10 mg/L and albumin ≥ 3.5 g/dL (1) CRP > 10 mg/L and albumin < 3.5 g/dL (2)
Prognostic nutritional index [10,12]	10 × serum albumin value (g/dl) + 0.005 × peripheral lymphocyte count (/uL) ≥ 45 (0) 10 × serum albumin value (g/dl) + 0.005 × peripheral lymphocyte count (/uL) < 45 (1)
Neutrophil-to-lymphocyte ratio [10,13]	Neutrophil count:/lymphocyte count
Platelet-to-lymphocyte ratio [10,14]	Platelet count/lymphocyte count
Lymphocyte-to-C-reactive protein ratio [15]	Lymphocyte count (10^9^/L)/C-reactive protein (mg/L) × 10^4^
Red blood cell distribution width	Standard deviation of red blood cell volume/mean cell volume × 100; higher values indicate greater variation in red blood cell size
Platelet distribution width	Measure of the variability in platelet size; higher values indicate greater variation in platelet size, associated with platelet activation

**Table 2 medicina-60-01191-t002:** Patients’ characteristics.

	Immunoglobulin A Nephropathy N = 196	Membranous Nephropathy N = 138	*p*
Age, years *	43 (41–46)	55 (51–58)	<0.001
Male sex, %	69	65	0.4
Clinical presentation, %			<0.001
Nephrotic syndrome	6	85	
Nephritic syndrome	85	10	
AUA	1	4	
AKI	1	0	
CKD	7	1	
Arterial hypertension, %	73	58	0.004
MAP, mmHg	100 (98.3–113.3)	100 (96.6–103.3)	0.1
Diabetes mellitus, %	8	13	0.2
Charlson comorbidity score	2 (2–3)	2 (2–3)	0.1
**Kidney function parameters**			
eGFR, mL/min	39.1 (35.1–42.9)	65.0 (62.4–71.2)	<0.001
Proteinuria, g/g	1.2 (1.1–1.5)	5.7 (4.7–6.6)	<0.001
Hematuria, h/HPF	180 (120–190)	20 (10–30)	<0.001
**Inflammation parameters**			
Hemoglobin, g/dL *	13.2 (12.7–13.5)	13.3 (12.8–13.5)	0.1
Neutrophil count, 10^3^/μL	4.8 (4.6–5)	4.5 (4.2–5.1)	0.3
Lymphocyte count, 10^3^/μL	2.05 (1.9–2.2)	2.1 (1.9–2.2)	0.5
Platelet count, 10^3^/μL	264 (255–274)	301.5 (279–320)	0.003
Erythrocyte sedimentation rate, mm/h	28.5 (24.0–34.0)	47.7 (40.0–60.0)	<0.001
Fibrinogen, mg/dL	470 (446–519)	758 (704–782)	<0.001
Serum albumin, g/dL	4.2 (4.1–4.3)	2.8 (2.6–3)	<0.001
C-reactive protein, mg/L	3 (2–3)	3 (2–3)	0.6
**Systemic-inflammation-based prognostic scores and hematological indices**			
Glasgow prognostic score			<0.001
0	78	19	
1	19	62	
2	3	19	
Modified Glasgow prognostic score			<0.001
0	85	82	
1	12	0	
2	3	18	
Prognostic nutritional index			<0.001
0	30	0	
1	70	100	
Neutrophil-to-lymphocyte ratio	2.3 (2.1–2.5)	2.1 (2.1–2.3)	0.1
Platelet-to-lymphocyte ratio	123.8 (118.8–130.8)	142.7 (130.3–155.0)	0.01
Lymphocyte-to-C-reactive protein ratio	7266.6 (6000.0–8500.0)	6500.0 (4285.7–9250.0)	0.8
Red blood cell distribution width	14.6 (14.4–14.8)	14.9 (14.8–15.2)	<0.001
Platelet distribution width	15.5 (15.5–15.6)	15.4 (15.4–15.5)	0.5
**Treatment**			
ACEI/ARB, n (%)	116 (59)	73 (53)	0.2
Immunosuppressive treatment, n (%)	92 (47)	110 (80)	<0.001
**Outcome**			
End-stage kidney disease, n (%)	72 (37)	17 (12)	<0.001
Death, n (%)	38 (19)	29 (21)	0.7

* mean. Continuous variables are presented as mean or median with 95% confidence interval (CI), categorical variables were presented as percentages. ACEI/ARB, angiotensin-converting enzyme inhibitor/angiotensin receptor blocker; AKI, acute kidney injury; AUA, asymptomatic urinary abnormality; CKD, chronic kidney disease; eGFR, estimated glomerular filtration rate; HPF, high power field; MAP, mean arterial blood pressure.

**Table 3 medicina-60-01191-t003:** Relationship between inflammation-based scores and hematological markers of inflammation with kidney survival.

	IgA Nephropathy				Membranous Nephropathy			
	Univariate		Multivariate *		Univariate		Multivariate *	
	HR (95%CI)	*p*	HR (95%CI)	*p*	HR (95%CI)	*p*	HR (95%CI)	*p*
Glasgow prognostic score		0.02		0.3		0.7		0.8
0	Reference		Reference		Reference		Reference	
1	1.98 (1.14–3.44)	0.01	0.89 (0.48–1.66)	0.7	1.60 (0.35–7.35)	0.5	1.40 (0.28–6.88)	0.6
2	2.27 (0.70–7.32)	0.1	0.41 (0.12–1.39)	0.1	2.01 (0.33–12.08)	0.4	1.67 (0.25–10.90)	0.5
Modified Glasgow prognostic score		0.3		0.1				
0	Reference		Reference		Reference		Reference	
1	1.37 (0.68–2.77)	0.3	0.54 (0.25–1.16)	0.1	-		-	
2	2.05 (0.64–6.57)	0.2	0.32 (0.10–1.20)	0.09	1.37 (0.38–4.89)	0.6	1.26 (0.35–4.54)	0.7
Prognostic nutritional index								
0	Reference		Reference		-		-	
1	2.29 (1.27–4.12)	0.005	1.13 (0.60–2.14)	0.6	0.96 (0.88–1.03) **	0.3	0.93 (0.84–1.02) **	0.1
Neutrophil-to-lymphocyte ratio	1.03 (0.89–1.20)	0.6	0.71 (0.55–0.90)	0.006	1.15 (0.92–1.42)	0.2	1.12 (0.81–1.55)	0.4
Platelet-to-lymphocyte ratio	0.99 (0.99–1.00)	0.7	1.00 (0.99–1.00)	0.3	1.00 (1.00–1.01)	0.06	1.00 (0.99–1.01)	0.2
Lymphocyte-to-C-reactive protein ratio	0.94 (0.77–1.15)	0.5	1.48 (1.19–1.83)	<0.001	0.92 (0.61–1.39)	0.7	1.07 (0.70–1.63)	0.7
Red blood cell distribution width	0.87 (0.70–1.08)	0.2	0.97 (0.76–1.24)	0.8	1.40 (0.82–2.40)	0.2	1.27 (0.72–2.22)	0.4
Platelet distribution width	0.90 (0.77–1.07)	0.2	1.16 (0.98–1.37)	0.06	0.78 (0.55–1.12)	0.1	0.6 (0.58–1.27)	0.4

* adjusted for eGFR, proteinuria g/g. ** as continuous variable, all patients had a score of 1 (see Table 1).

**Table 4 medicina-60-01191-t004:** Relationship between inflammation-based scores and hematological markers of inflammation with survival.

	IgA Nephropathy				Membranous Nephropathy			
	Univariate		Multivariate *		Univariate		Multivariate *	
	HR (95%CI)	*p*	HR (95%CI)	*p*	HR (95%CI)	*p*	HR (95%CI)	*p*
Glasgow prognostic score		<0.001		0.09		0.2		0.2
0	Reference		Reference		Reference		Reference	
1	4.46 (2.21–8.98)	<0.001	2.43 (1.04–5.64)	0.03	1.40 (0.47–4.17)	0.5	1.17 (0.33–4.08)	0.8
2	4.81 (1.40–16.49)	0.01	1.00 (0.26–3.76)	0.9	2.52 (0.75–8.40)	0.1	2.30 (0.59–8.97)	0.2
Modified Glasgow prognostic score		<0.001		0.1				
0	Reference		Reference		Reference		Reference	
1	3.92 (1.85–8.31)	<0.001	2.16 (0.95–4.89)	0.06	-		-	
2	3.97 (1.18–13.30)	0.02	1.15 (0.28–4.64)	0.8	1.94 (0.85–4.38)	0.1	1.42 (0.92–2.20)	0.1
Prognostic nutritional index								
0	Reference		Reference		-		-	
1	2.98 (1.16–7.67)	0.02	1.03 (1.00–1.06)	0.4	0.93 (0.87–0.99) **	0.02	0.94 (0.87–1.01) **	0.1
Neutrophil-to-lymphocyte ratio	1.10 (0.91–1.33)	0.3	0.90 (0.65–1.23)	0.5	1.25 (1.10–1.41)	<0.001	1.35 (1.11–1.64)	0.002
Platelet-to-lymphocyte ratio	0.99 (0.98–1.00)	0.04	0.99 (0.98–1.00)	0.04	1.005 (1.001–1.010)	0.01	1.00 (1.00–1.01)	0.01
Lymphocyte-to-C-reactive protein ratio	0.81 (0.62–1.05)	0.1	1.20 (0.89–1.61)	0.2	0.74 (0.59–0.96)	0.02	0.83 (0.61–1.13)	0.2
Red blood cell distribution width	1.10 (0.91–1.32)	0.3	1.23 (0.90–1.69)	0.1	1.78 (1.18–2.69)	0.006	1.76 (1.06–2.90)	0.02
Platelet distribution width	0.81 (0.66–1.00)	0.05	1.03 (0.83–1.29)	0.7	0.94 (0.68–1.30)	0.7	1.14 (0.81–1.61)	0.4

* adjusted for age, Charlson comorbidity score, eGFR, proteinuria g/g. ** as continuous variable, all patients had a score of 1 (see Table 1).

## Data Availability

The data underlying this article will be shared upon reasonable request to the corresponding author.

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
