# Peer review of "Prognostic Value of Inflammation Scores and Hematological Indices in IgA and Membranous Nephropathies: An Exploratory Study"

_medicina, 2024, doi:10.3390/medicina60081191_

Round 1

Reviewer 1 Report

Comments and Suggestions for Authors

English is fine. Materials and methods are robust. 

Have you found possible associations between inflammatory indices and cancer risks in these patients? You can refer to these articles:

  • 10.3390/cancers16030651

  •  

Author Response

English is fine. Materials and methods are robust. 

Have you found possible associations between inflammatory indices and cancer risks in these patients? You can refer to these articles: 10.3390/cancers16030651

Thank you for your positive feedback on the English quality and the robustness of our materials and methods.

Regarding your inquiry about possible associations between inflammatory indices and cancer risks in these patients, we regret to inform you that we did not have the necessary data to conduct this specific analysis. Our dataset did not include comprehensive information on cancer diagnoses or related variables, which would be required to explore such associations meaningfully.

We appreciate your suggestion and the reference to relevant literature. Future studies could certainly benefit from incorporating cancer risk data to investigate potential links between inflammatory indices and cancer in patients with glomerular diseases.

Thank you once again for your insightful comments and suggestions.

Reviewer 2 Report

Comments and Suggestions for Authors

Dear Editors,

The study titled "Prognostic Value of Inflammation Scores and Haematological Indices in IgA and Membranous Nephropathies: An Exploratory Study" offers a comprehensive analysis of the prognostic utility of various systemic inflammation-based scores and hematological indices in patients with IgA Nephropathy (IgAN) and Membranous Nephropathy (MN). The objective of the research is to investigate the association between the described biomarkers and major clinical outcomes, ESKD, and mortality within a group of 334 patients originating from a single tertiary center.

There are some concerns that need to be addressed.

-        The abstract is hard to follow and needs to be more succinct to better showcase the results.

-        The authors are advised to include some statistics on IgAN and MN with a brief description of their respective pathophysiologies within the introduction section, as well as the existing markers, whether used in clinical practice or not.

-        While the chosen confounders are pertinent, other potential confounders (e.g., medication adherence, lifestyle factors (e.g., diet, physical activity) are not accounted for, which could influence the study outcomes.

-        There is no explicit mention of tests for multicollinearity among the variables included in the Cox models. Multicollinearity can inflate the variances of the estimated coefficients, leading to less reliable results. It would be prudent to check for this, using the Variance Inflation Factor for example.

-        In table 1, inflammation-based scores have to be referenced using additional literature.

Author Response

Dear Editors,

The study titled "Prognostic Value of Inflammation Scores and Haematological Indices in IgA and Membranous Nephropathies: An Exploratory Study" offers a comprehensive analysis of the prognostic utility of various systemic inflammation-based scores and hematological indices in patients with IgA Nephropathy (IgAN) and Membranous Nephropathy (MN). The objective of the research is to investigate the association between the described biomarkers and major clinical outcomes, ESKD, and mortality within a group of 334 patients originating from a single tertiary center.

There are some concerns that need to be addressed.

  • The abstract is hard to follow and needs to be more succinct to better showcase the results.

We have reformulated the abstract.

  • The authors are advised to include some statistics on IgAN and MN with a brief description of their respective pathophysiologies within the introduction section, as well as the existing markers, whether used in clinical practice or not.

We have included in the introduction section the informations requested by the reviewer.

IgAN is characterized by the deposition of IgA immune complexes in the glomeruli, leading to local inflammation and glomerular injury. This condition often presents with hematuria and varying degrees of proteinuria. Current markers for IgAN include serum creatinine, proteinuria levels, and eGFR, though these are not specific to the disease's unique immune-mediated mechanisms. MN involves the deposition of immune complexes on the glomerular basement membrane, causing thickening and damage to the glomeruli. Patients typically present with heavy proteinuria and nephrotic syndrome. Established markers include anti-PLA2R antibodies and proteinuria levels, which are used in clinical practice to monitor disease activity and response to therapy.

  • While the chosen confounders are pertinent, other potential confounders (e.g., medication adherence, lifestyle factors (e.g., diet, physical activity) are not accounted for, which could influence the study outcomes.

We agree with the reviewer’s comment and have added this as a limitation to our study: While the chosen confounders are pertinent, other potential confounders (e.g., medication adherence, lifestyle factors such as diet and physical activity) are not accounted for, which could influence the study outcomes.

  • There is no explicit mention of tests for multicollinearity among the variables included in the Cox models. Multicollinearity can inflate the variances of the estimated coefficients, leading to less reliable results. It would be prudent to check for this, using the Variance Inflation Factor for example.

We employed two methods to test for collinearity among our predictor variables: (i) the variance inflation factor (VIF), with a desirable threshold of VIF <10, and (ii) the absolute value of correlation coefficients, aiming for |r| or |rs| <0.7. Our analysis indicated no significant collinearity among the variables used in the Cox proportional hazard models.

  • In table 1, inflammation-based scores have to be referenced using additional literature.

We added additional references to Table 1.

Reviewer 3 Report

Comments and Suggestions for Authors

While the article is interesting, it is imprecise in many places.

Abstract:
It lacks a clearly defined goal and conclusions.
Instead of conclusions, there is an entry here that is just a summary - a presentation of the results: "In MN, NLR, PLR and RDW are predictors of mortality but not kidney survival. Future studies with larger cohorts are necessary to validate these markers and understand their mechanisms."

What mechanisms do the authors have in mind in the context of markers?

I was wondering what the advantage of this article is over similar ones already published.

Although the researchers wrote in the limitations of the study: "Secondly, we only analyzed patients with IgAN and MN at the time of presentation, i.e., at the time of kidney biopsy. This means our findings do not consider potential changes in disease status or treatment responses over time, which could affect long-term outcomes."

I wonder how NLR, PLR, and RDW can be indicators of mortality since they were analyzed only at one-time point; they could have changed many times. Other factors may have occurred during the course of the disease. The treatment method also changed during the analyzed period.

The authors must address this comment. I am not sure that writing about it in the Limitations of the Study solved the issue.

It would be useful to have a research diagram to understand exactly what was analyzed and when. I suggest that the authors add it.
From technical issues:

Table 2 lacks information on how the results were presented - whether they are medians or means, and what the entry in brackets means, which should be explained below the table.
In the case of Treatment - should find the entry ACEI/ARB, n(%)

Conclusions in the article, similar to those in the abstract, should be reformulated.

Language: Tenses used are different. This should be changed.

Comments on the Quality of English Language

Moderate editing of the English language is required.

Author Response

While the article is interesting, it is imprecise in many places.

Abstract:
It lacks a clearly defined goal and conclusions.
Instead of conclusions, there is an entry here that is just a summary - a presentation of the results: "In MN, NLR, PLR and RDW are predictors of mortality but not kidney survival. Future studies with larger cohorts are necessary to validate these markers and understand their mechanisms."

We have revised the abstract to enhance clarity.

Background and Objectives: Systemic inflammation-based prognostic scores and hematological indices have shown value in predicting outcomes in various clinical settings. However, their effectiveness in predicting outcomes specifically for IgA nephropathy (IgAN) and membranous nephropathy (MN), the most common primary glomerular diseases diagnosed by kidney biopsy, has not been thoroughly investigated. Materials and Methods: We conducted a retrospective, observational study involving 334 adult patients with biopsy-proven IgAN (196 patients) and MN (138 patients) from January 2008 to December 2017 at a tertiary center. We assessed six prognostic scores [Glasgow Prognostic Score (GPS), modified GPS (mGPS), Prognostic Nutritional Index (PNI), Neutrophil-to-Lymphocyte Ratio (NLR), Platelet-to-Lymphocyte Ratio (PLR), Lymphocyte-to-C-Reactive Protein Ratio (LCRP)] and two hematological indices [Red Blood Cell Distribution Width (RDW), Platelet Distribution Width (PDW)] at diagnosis and examined their relationship with kidney and patient survival. Results: End-stage kidney disease (ESKD) occurred more frequently in the IgAN group compared to the MN group (37% vs. 12%, p = 0.001). The mean kidney survival time was 10.7 years in the IgAN cohort and 13.8 years in the MN cohort. After adjusting for eGFR and proteinuria, lower NLR and higher LCRP were significant risk factors for ESKD in IgAN. In the MN cohort, no systemic inflammation-based scores or hematological indices were associated with kidney survival. There were 38 deaths (19%) in the IgAN group and 29 deaths (21%) in the MN group, showing no significant difference in mortality rates. The mean survival time was 13.4 years for IgAN and 12.7 years for MN. In IgAN, lower PLR was associated with higher mortality after adjusting for age, Charlson comorbidity score, eGFR, and proteinuria. In MN patients, higher NLR, PLR, and RDW were associated with increased mortality. Conclusions: NLR and LCRP are significant predictors for ESKD in IgAN, while PLR is linked to increased mortality. In MN, NLR, PLR, and RDW are predictors of mortality but not kidney survival. These findings underscore the need for disease-specific biomarkers and indicate that systemic inflammatory responses play varying roles in the progression and outcomes of these glomerular diseases. Future studies with larger cohorts are necessary to validate these markers and understand their mechanisms.

What mechanisms do the authors have in mind in the context of markers?

Kidney survival in IgAN

In our multivariate analysis adjusted for eGFR and proteinuria, a lower NLR emerged as an independent risk factor for ESKD. These findings could be explained by the lower eGFR at diagnosis in our population. In this context, a lower NLR might reflect chronic inflammation, immune exhaustion, and malnutrition. Thus, in earlier stages of IgAN, a higher NLR could reflect the "disease-specific" immune-mediated nephron loss, while in advanced stages, a lower NLR might reflect the generic response to nephron loss (id est, CKD progression). While higher NLR typically reflects acute inflammation (high neutrophil counts), lower NLR might reflect a shift towards chronic inflammation and immune dysregulation, which are detrimental in long-term IgAN progression. This scenario is consistent with the finding that a higher LCRP (higher lymphocyte counts relative to CRP) was also associated with ESKD.

Kidney survival in MN

MN is primarily characterized by immune complex deposition in the glomeruli, leading to podocyte damage and proteinuria, which may not be directly reflected in the systemic inflammatory markers measured. Moreover, the progression of MN can be heavily influenced by other factors such as the degree of proteinuria, response to immunosup-pressive therapy, and underlying comorbidities like hypertension and diabetes. These factors may have a more significant impact on renal outcomes in MN than the systemic inflammatory response, which is why the prognostic scores and hematological indices based on inflammation did not show a significant association with ESKD in this patient group

Patient survival in IgAN

IgAN is characterized by chronic immune-mediated inflammation, where lower PLR values indicate severe disease and immune dysregulation. This is because IgAN involves the deposition of IgA immune complexes in the glomeruli, leading to local inflammation and increased lymphocyte activity, which can reduce PLR.

Patient survival in MN

MN is driven by autoimmune processes targeting glomerular podocytes, resulting in a pro-thrombotic state. Higher PLR values in MN reflect increased platelet activation and aggregation, which are associated with a higher risk of thrombosis and systemic in-flammation, thereby increasing mortality risk.

NLR was associated with mortality in MN patients. Studies suggest that both a high baseline NLR and increases in NLR over time are linked to higher mortality rates. The strong association between high NLR and low serum albumin levels further supports NLR as a reliable mortality marker. Therefore, in MN patients presenting with nephrotic syndrome, NLR may have greater predictive utility than al-bumin because NLR increases in the blood much faster (6–8 hours) compared to the slower decrease in albumin levels (19–21 days).

I was wondering what the advantage of this article is over similar ones already published.

Our study has several notable advantages:

  1. It focuses on the two most common primary glomerular diseases, IgAN and MN, which is significant from both clinical and practical perspectives.
  2. It includes the largest number of systemic inflammation-based prognostic scores and hematological indices studied in IgAN and MN to date.
  3. It utilizes hard endpoints, specifically end-stage kidney disease and mortality.
  4. It involves long-term follow-up, which is appropriate for these chronic diseases.

Although the researchers wrote in the limitations of the study: "Secondly, we only analyzed patients with IgAN and MN at the time of presentation, i.e., at the time of kidney biopsy. This means our findings do not consider potential changes in disease status or treatment responses over time, which could affect long-term outcomes."
I wonder how NLR, PLR, and RDW can be indicators of mortality since they were analyzed only at one-time point; they could have changed many times. Other factors may have occurred during the course of the disease. The treatment method also changed during the analyzed period.

The authors must address this comment. I am not sure that writing about it in the Limitations of the Study solved the issue.

Thank you for your constructive comment.

Our study collected data at the time of kidney biopsy, representing the moment of diagnosis. This critical point is essential for assessing prognosis and guiding initial treatment strategies. While NLR, PLR, and RDW can change over time and be influenced by treatments and disease progression, our goal was to evaluate their prognostic value at this pivotal initial time point.

We acknowledge the limitation of analyzing these markers only at diagnosis. This does not account for changes over the course of the disease or different treatment impacts. However, as an exploratory study with long-term follow-up, we aim to identify initial prognostic indicators that can help guide early treatment decisions and predict long-term outcomes.

Future studies should include multiple time points to better understand the dynamic nature of these markers and their relationship with long-term outcomes. This will help develop more robust predictive models that incorporate changes over time.

It would be useful to have a research diagram to understand exactly what was analyzed and when. I suggest that the authors add it.

Inflammation-based scores and hematological markers of inflammation were analyzed at the time of kidney biopsy, which is the moment of diagnosis. These definitions are clearly detailed in Table 1, making a diagram redundant. However, if the editor and reviewer deem the diagram necessary, we will include it in the manuscript.

From technical issues:
Table 2 lacks information on how the results were presented - whether they are medians or means, and what the entry in brackets means, which should be explained below the table.
In the case of Treatment - should find the entry ACEI/ARB, n (%).

As the reviewer suggested, we added the mising data.

Conclusions in the article, similar to those in the abstract, should be reformulated. 

The conclusions section of the manuscript was reformulated:

This study highlights the importance of systemic inflammation-based scores in predicting outcomes in IgAN and MN. In IgAN, NLR and LCRP are significant predictors for ESKD progression, while PLR is associated with increased mortality. In MN, NLR, PLR, and RDW predict mortality but not kidney survival. These findings underscore the necessity for disease-specific biomarkers and indicate that systemic inflammatory responses influence the progression and outcomes of these glomerular diseases differently. Future studies with larger cohorts are essential to validate these markers and better understand their mechanisms.

Language: Tenses used are different. This should be changed.

Comments on the Quality of English Language

Moderate editing of the English language is required.

 We have revised the manuscript for English quality.

Round 2

Reviewer 3 Report

Comments and Suggestions for Authors

The authors have significantly improved the article, but one issue still concerns me.

Last sentence in abstract: "Future studies with larger cohorts are necessary to validate these markers and understand their mechanisms". There is a similar sentence in Conclusions.

Markers are not involved in mechanisms. Markers may be subject to changes under the influence of various mechanisms. Please correct this.

Comments on the Quality of English Language

Minor editing of the English language is still required.

Author Response

The authors have significantly improved the article, but one issue still concerns me.

Last sentence in abstract: "Future studies with larger cohorts are necessary to validate these markers and understand their mechanisms". There is a similar sentence in Conclusions.

Markers are not involved in mechanisms. Markers may be subject to changes under the influence of various mechanisms. Please correct this.

Dear Reviewer,

Thank you for your positive feedback and for pointing out the issue regarding the last sentence in the abstract and conclusions.

We agree with your observation and have revised the sentence to reflect that markers are influenced by various mechanisms rather than being directly involved in them. The corrected sentences are as follows:

Abstract:

"Future studies with larger cohorts are necessary to validate these markers."

Conclusions:

"Future studies with larger cohorts are essential to validate these markers and investigate the mechanisms that affect them."

We appreciate your thorough review and constructive feedback, which have helped improve the clarity and accuracy of our manuscript.